# Changes in Sensitization Patterns in the Last 25 Years in 619 Patients with Confirmed Diagnoses of Immediate Hypersensitivity Reactions to Beta-Lactams

**DOI:** 10.3390/biomedicines10071535

**Published:** 2022-06-28

**Authors:** María del Valle Campanón Toro, Esther Moreno Rodilla, Alicia Gallardo Higueras, Elena Laffond Yges, Francisco J. Muñoz Bellido, María Teresa Gracia Bara, Cristina Martin García, Vidal Moreno Rodilla, Eva M. Macías Iglesias, Sonia Arriba Méndez, Miriam Sobrino García, Ignacio Dávila

**Affiliations:** 1Allergy Service, University Hospital of Salamanca, 37007 Salamanca, Spain; mvallect@gmail.com (M.d.V.C.T.); agallardoh@saludcastillayleon.es (A.G.H.); mlaffond@saludcastillayleon.es (E.L.Y.); fjmbellido@saludcastillayleon.es (F.J.M.B.); mtgracia@saludcastillayleon.es (M.T.G.B.); cmartingarci@saludcastillayleon.es (C.M.G.); emacias@saludcastillayleon.es (E.M.M.I.); sdearriba@saludcastillayleon.es (S.A.M.); miriamsobrino@saludcastillayleon.es (M.S.G.); idg@usal.es (I.D.); 2Allergy Service, Llerena-Zafra Hospital, 06300 Zafra, Spain; 3IBSAL (Institute for Biomedical Research of Salamanca), 37007 Salamanca, Spain; 4Department of Biomedical and Diagnostic Sciences, Salamanca Medical School, University of Salamanca, 37007 Salamanca, Spain; 5RETIC de Asma, Reacciones Adversas y Alérgicas (ARADYAL), 28040 Madrid, Spain; 6Department of Computer Science and Automatic, University of Salamanca, 37008 Salamanca, Spain; vmoreno@usal.es

**Keywords:** beta-lactam, drug allergy, skin tests, immediate hypersensitivity reactions, sensitization patterns, drug provocation test

## Abstract

Beta-lactam (BL) drugs are the antibiotics most prescribed worldwide due to their broad spectrum of action. They are also the most frequently implied in hypersensitivity reactions with a known specific immunological mechanism. Since the commercialization of benzylpenicillin, allergic reactions have been described; over the years, other new BL drugs provided alternative treatments to penicillin, and amoxicillin is now the most prescribed BL in Europe. Diagnosis of BL allergy is mainly based on skin tests and drug provocation tests, defining different sensitization patterns or phenotypes. In this study, we evaluated 619 patients with a confirmed diagnosis of BL-immediate allergy during the last 25 years, using the same diagnostic procedures with minor adaptations to the successive guidelines. The initial eliciting drug was benzylpenicillin, which changed to amoxicillin with or without clavulanic acid and cephalosporins in recent years. In skin tests, we found a decrease in sensitivity to major and minor penicillin determinants and an increase in sensitivity to amoxicillin and others; this might reflect that the changes in prescription could have influenced the sensitization patterns, thus increasing the incidence of specific reactions to side-chain selective reactions.

## 1. Introduction

Beta-lactam (BLs) antibiotics are one of the drugs most frequently prescribed for infections and the most common cause of drug-induced immediate hypersensitivity reactions (HSR) [1,2]. Since the commercialization of benzylpenicillin (BP), it has been widely used; over time, the development of new BLs provided therapeutic alternatives in infections, and amoxicillin (AX) is now the drug most prescribed in Europe [3,4].

BP and ampicillin (AMP) were the first BLs, marketed in the 1950 and 1960s, respectively [5] and the first causing HSRs, as they were the most used antibiotics in Europe and also in the United States until the 1980s [6,7,8]. From 1975 when it first appeared, AX had begun to be prescribed due to its easy administration; thus it became the most frequently administered antibiotic, either alone or in combination with clavulanic acid (CLV), and displaced BP as the most widely used [6]. Additionally, a diversity of cephalosporins started to become an alternative treatment in several indications, mainly against resistant microorganisms and changing trends in antibiotics prescriptions [9].

The first published BL allergy studies showed similar sensitization profiles, with a high percentage of positive skin test (STs) results to major and minor BP determinants (Penicilloyl poly-l-lysine (PPL) and minor determinant mixture (MDM)) [10,11]. These patients were commonly sensitized to the BL ring and needed to avoid all BLs. Several years after introducing the new penicillin and cephalosporins to the market, changes in sensitization profiles were detected, and a higher heterogeneity in ST results was found. In addition to sensitization to the BL ring, there were also responses due to similarity in the R1 side chains and selective reactions to different BLs [12,13,14]. Such different sensitization profiles made it advisable to test a higher number of determinants to achieve an adequate diagnosis and to advise for future treatment. Studies adding AX to STs in diagnosing BL allergy found it the most frequently positive BL [13,15,16], with selective responses in a significant percentage of cases [17,18]. Moreover, testing CLV and different cephalosporins has become essential in diagnosing patients with allergic reactions because of increased ST sensitivity [16,19,20,21].

Drug prescription trends, and consequently, ST results, and profiles vary among countries [22,23]. Thus, the frequency of use of each drug is related to the frequency of causing HSRs and determines different sensitization profiles or phenotypes [8,18]. Patients with immediate reactions to BLs can be allergic to several antibiotics, a subgroup of drugs with side-chain similarities, or just a single drug. Nonetheless, avoiding all BLs when penicillin or cephalosporin allergy is suspected remains a prevalent practice despite the study results of sensitization profiles. Avoiding all BLs represents a significant public health issue [24,25,26,27] with well-documented consequences, including greater use of second-line antibiotics, increasing antibiotic-resistant infections, more adverse effects, more extended hospitalization, hospital re-admissions, and higher costs. An accurate study of reactions and sensitization profiles allows avoidance of allergy reactions due to cross-reactivity or too broad avoidance recommendations, ensuring an optimized use of the medicines [28,29,30,31].

Even though new diagnostic methods are gaining importance in the diagnosis of drug HSRs [2,8,27,29,32], not only by the development of different in vitro studies but also by predicting models, STs are still one of the most reliable tools to diagnose patients in daily practice [14,29,33,34]. Since the sensitivity and specificity of STs are suboptimal, drug provocation tests (DPTs) are also needed to achieve diagnosis in many cases [28,29]. This study aims to describe the different sensitization profiles in a large group of patients using ST results and DPTs and analyze whether the changing trends in drug prescription have influenced and changed sensitization profiles.

## 2. Material and Methods

We retrospectively searched our database for all patients older than 14 years who were evaluated in our allergy department at a Spanish public hospital with suspected BL allergy between 1995 and 2020. We collected those with a confirmed diagnosis of immediate hypersensitivity to one or more BL in database performed with SPSS software, version 26 (IBM corporation, Armonk, NY, USA). Patients were included if they suffered an immediate reaction after BL administration, which occurred in the first 6 h after drug administration. Only reactions characterized by urticaria, angioedema (AE), or both, or anaphylaxis were included, the latter being diagnosed according to criteria proposed by Sampson et al. [35]. The study was performed following the requirements expressed in the Declaration of Helsinki and the current Spanish legislation on the conduct of observational studies (Royal Decree 957/2020, 3 November) and the Law 14/2007 on Biomedical Research.

### 2.1. Allergy Workup

Diagnosis of BL HSRs has been protocolized in our allergy department since 1995 and adapted to successive guidelines [34,36,37]. First, a thorough clinical history was conducted on patients, collecting a detailed description of symptoms and circumstances of the reaction, implied drug, date of reaction or reactions, and other possible similar drugs tolerated after the reaction. A blood sample was drawn when the allergist in charge of the patient so decided. In vivo studies began with STs to different reagents (see below in Skin Tests). If STs were negative, patients underwent controlled DPTs. If the clinical history highly suggested HSR and the reaction occurred more than six months before the study, STs and DPTs were repeated after three weeks. Exclusion criteria were pregnancy, beta-blocker drug use, and severe renal, cardiovascular, or respiratory disease. Written informed consent was obtained from all patients before starting the allergy workup. For minor patients, written informed consent was obtained from both the parents and the minor.

### 2.2. Detection of Total and Specific IgE in Serum

Serum total IgE and specific IgE assays with penicilloyl G, penicilloyl V, ampicilloyl, amoxicilloyl, and cefaclor using ImmunoCAP (Phadia, Uppsala, Sweden, now Thermo Fisher Scientific) were performed when the allergy specialist in charge of the patient so considered. A value of 0.35 kUA/L or greater was considered positive.

### 2.3. Skin Tests (STs)

All STs were performed as previously described [38], using PPL (Penicilloyl poly-L-lysine) and MDM (minor determinant mixture) (both from Allergopen; Allergopharma-JGKG, Reinbek, Germany) until 2006; and BP-OL (0.04 mg/mL) and MD (minor determinant) (0.5 mg/mL) (both, respectively, Benzylpenicilloyl-octa-l-lysine and benzylpenilloate, from DAP- Diater, Madrid, Spain) were performed afterward. BP 10,000 UI and AX 20 mg/mL were tested as well. After 2006, we also included cefuroxime 2mg/mL and meropenem (1 mg/mL) [29,37]. STs with other cephalosporins with similar and different side chains were performed on several patients. After 2013, as recommended by [37] in the case of cephalosporins, the concentration of 20 mg/mL also was tested if STs with 2 mg/mL were negative. For BL, other than the ones mentioned above, the concentrations used were those referred to in the bibliography.

First, reagents were skin tested by prick (SPT) on the volar forearm and considered positive when a wheal larger than 3 mm diameter with surrounding erythema was present after 20 min. If negative, intradermal tests (IDTs) were performed and considered positive if the initial wheal diameter increased 3 mm or more with surrounding erythema after 20 min. Positive results in one or more reagents were considered diagnostic [29].

### 2.4. Drug Provocation Tests (DPTs)

If skin tests were negative, patients underwent a DPT with the suspected BL in a single-blind placebo-controlled challenge. Increasing doses (1/8-1/4-1/2-1/2 of a full dose), according to the local protocol DPT, were administered at 45 min intervals [29,30].

### 2.5. Re-Evaluation of Negative Patients

When STS and DPT with the eliciting BL were negative, and reaction occurred more than six months before the study, the study was repeated three weeks later using the same protocol.

### 2.6. Statistical Analyses

Clinical data of patients, information about reaction and drug implications, and diagnostic performance results were collected. The normal distribution of the results was checked by the Kolmogorov–Smirnov test, with the result that our sample of patients does not follow a normal distribution. Continuous data were summarized as mean, median, and standard deviations, and categorical data were expressed in count values (%). The sample was divided into four groups according to quartiles relative to the reaction date.

Statistical analyses and quartiles were performed using SPSS software, version 26 (IBM corporation, Armonk, NY, USA). Correlation comparisons were made with Fisher’s exact test. Statistical significance was defined as *p* < 0.05.

## 3. Results

Between 1995 and 2020, 619 patients were diagnosed with immediate hypersensitivity to BLs. The sample was divided into four quartiles according to the date of the reaction. The groups were as follows: Q1: 1950–1995 (144 patients); Q2: 1996–2004 (165 patients); Q3: 2005–2011 (144 patients); and Q4: 2012–2020 (163 patients). Characteristics of patients and reactions are shown in Table 1.

The mean age was 46.57 years old, with 345 females (55.7%) and 82 patients (13.5%) having atopy antecedents. There were notable differences among quartiles regarding the time between reaction and study, with a global mean of 47.20 ± 99.06 months. Concerning the types of reactions, 366 (59.4%) patients suffered from urticaria and/or AE, and 250 (40.6%) patients had anaphylaxis following BL administration. AX and AX-CLV were the most frequently involved BLs (39.6% and 30.4%, respectively), followed by cephalosporins (15.5%) and BP (8.9%).

It is noticeable how the proportion of reactions due to BP reduced over the years. Thus, it was the second drug most frequently involved in Q1 (33.3%), but its proportion drastically decreased to the last position in Q4 (0). The opposite trend was observed with AX, AX-CLV, and cephalosporins, which were more frequently implied from Q2 onward; AX and AX-CLV were most frequently used eliciting drugs in all groups, representing 70% of the total. Cephalosporins also considerably increased, varying from 7.7% of patients in Q1 to 23.9% in Q4.

Concerning diagnostic procedures, most patients were diagnosed by SPTs (17%) or IDTs (62.2%); DPTs were necessary for 9.6% of patients. In the re-evaluation study, SPTs were positive in 1.5% of total patients, IDTs in 7%, and DPTs in 0.3%. Only 2.4% of patients were diagnosed with a positive specific IgE.

### 3.1. Skin Tests Results

Regarding STs, penicillin reagents (PPL/MDM/BP) were positive in 215 patients of 612 tested (35.1% of total) and amoxicillin (AX) in 292 patients of 608 tested (48% of total). (Table 2).

When comparing quartiles, significant differences were found. Penicillin reagents were positive in 57.6% of patients in Q1, but the proportion progressively descended. Thus, in Q2, the proportion was 30.3%, and in Q3 and Q4, it was 32.9% and 22.1%, respectively. A significant correlation between quartiles and ST results was found (Fisher’s test < 0.001) (Figure 1).

For AX ST results, Q1 had 31.3% positive STs, which was significantly lower than the proportion found in other quartiles (53.9% in Q2, 52.5% in Q3, and 52.7% in Q4). Therefore, a significant correlation between quartiles and STs results was found (Fisher’s test < 0.001) (Figure 2).

Other BLs also were tested in 182 patients (29.4% of patients), yielding positive results in 152 (24.7%). The percentage of positive STs was 12.5% for Q1, whereas Q2, Q3, and Q4 showed similar proportions (29.1%, 24.3%, and 24.7%, respectively). It should be noted that the progressive increase in the number of patients who were tested with other BLs required years to be diagnosed, from 18 in Q1 to 52 in Q4. When comparing Q1 with the rest of the quartiles, a significant difference was found (Fisher’s test *p* < 0.0001) (Figure 3).

Focusing on AX and AX-CLV, representing 39.6% and 30.4% of total patients, 240 patients had positive STs exclusively to AX (38.7% of total patients). Analyzing the differences by quartiles, in line with the increased frequency of AX as a triggering drug over the years, we found that 34 patients in Q1 (23.6%) had selective positive STs to AX. This percentage increased from Q2 onward, reaching 43.6%, 42.4%, and 43.6% in Q2, Q3, and Q4.

Concerning CLV, 30 patients had positive STs exclusively with CLV, being negative with AX and BP reagents; thus, CLV was the drug involved in 16% of patients who had reactions with AX-CLV, representing 4.8% of the global sample. Our first patient with hypersensitivity to CLV suffered the reaction in 1994, and since then, the frequency has increased over the years; there were seven patients (4.2%) in Q2, ten patients (6.9%) in Q3, and 12 patients (7.4%) in Q4 that were diagnosed with selective CLV hypersensitivity reactions.

When we correlated positive skin test results to different reagents and antibiotics-implied reactions, we found a statistically significant correlation between positive skin tests with PPL/MDM/BP or AX and reactions related to the administration of BP and amoxicillin, respectively (*p* < 0.0001). Patients presented positive STs to penicillin reagents (PPL/MDM/BP) in 76.3% of patients who suffered reactions with BP and positive STs to AX in 62.6% of patients in which AX or AX/CLV was the implied drug. (Table 3).

Analyzing exclusively cephalosporin-positive patients, we observed high percentages of positive STs only for the culprit or related cephalosporins: 76% for first-generation cephalosporins, 84.6% for second-generation, and 75% for third-generation (Table 2). Analyzing the whole set of cephalosporins without classifying them by generation, we found that 57.3% of patients had negative STs to AX and BP reagents. This fact was particularly relevant for second-generation cephalosporins, with 72.5% of patients having positive STs just to the involved drug and negative to penicillin and AX reagents.

Finally, in the case of patients with positive STs with cloxacillin, 82.8% of them had positive STs only with cloxacillin, testing negative to the rest of the reagents. In the case of patients allergic to PPZ-TZB, two of seven had positive STs for PPZ-TZB but negative for other reagents (28.6%).

### 3.2. Results of Drug Provocation Tests

DPTs were necessary to achieve diagnosis in 61 patients (9.8%); 11 patients in Q1 (7.6%), 13 in Q2 (7.9%), 13 in Q3 (9%), and 24 in Q4 (14.7%). There were no significant differences among quartiles (*p* = 0.1), but a tendency to increase was observed. According to the culprit drug, DPTs were more frequently needed in patients with hypersensitivity to AX and AX-CLV (3.55% and 2.26%, respectively, of the total patients). In addition, considering all patients with hypersensitivity to cephalosporins, DPTs were required to diagnose in 3.07% of the total sample, the third in frequency (Table 2).

The reactions registered during DPTs were usually mild. Most patients presented only with skin affectation (88.5%), U/AE the most frequent (78.7%), followed by other skin symptoms, such as unspecific rashes or itchy skin. Only four patients suffered anaphylactic reactions during DPTs, which represents 6.5% of patients diagnosed by DPTs and −0.65% of total patients with confirmed hypersensitivity; two patients had cough and urticaria, another one had urticaria and dyspnea, and the last one had hypotension and recovered within hours after routine treatment. None of these patients required intensive treatment or hospitalization due to reactions. There was no difference in the severity of reactions by patients previously referred to the study and those during DPTs.

### 3.3. Re-Evaluation of Negative Patients

Fifty-four patients (8.7% of 619 patients) were diagnosed during re-evaluation. The largest group belonged to Q1 (17.4%). In quartiles 2, 3, and 4, a positive re-evaluation was less frequently required (8.5%, 6.3%, and 3.7% of their patients, respectively). AX and AX-CLV were the most frequently involved drugs (44.4% and 29.6%, respectively), followed by BP (11.1%). When comparing groups and implied drugs, there were significant differences between Q1 and the rest of the quartiles aggregated as a group (Fisher’s test *p* < 0.01).

## 4. Discussion

This study analyzes an extensive series of 619 patients with a confirmed diagnosis of immediate hypersensitivity to BLs over a twenty-five years period. We found that the sensitization pattern notably changed over the 25 years.

Concerning drugs involved in the reactions, we globally found AX and AX-CLV as the BLs most frequently involved in reactions (69.9% of patients), with cephalosporins in second place (15.5%), followed by BP (8.9%), cloxacillin, and PPZ-TZB. Cloxacillin reactions also were found in 2.8% of the patients.

These frequencies are similar to those of other case series described in European populations [5,12,16].

In 2001, Torres et al. studied 290 patients with confirmed immediate hypersensitivity to penicillin. They compared two five-year periods between 1985 and 1995; in the second lustrum, they observed an increase in AX as a culprit drug (77.7% of cases) and a decrease in the frequency of BP from 6.2 to 1.6% [22]. In our series, BP represented 33.3% of cases in Q1 (1950–95), with a progressive decrease in the following years, reaching 0 in Q4 (years 2012–2020). In a child prospective cohort studied between 1990 and 2009, it was found similar frequencies, with AX in 64.9% of cases and cephalosporins in 21.5% [39]. Schrüfer et al. [40] recently found a higher frequency of cephalosporins, representing 84.3% of cases in a German center. 

Comparing the first patients studied in the 1990s with patients evaluated in recent years, changes in BLs involved in HSRs are appreciable. Thus, BPs have been moved from the first position in Q1 to last place in Q4, where no patient had received it. This change is notable from 1995 onward when AX became the most relevant drug, alone or with CLV. These data are related to changes in BL prescription habits. Thus, McCaig et al. [6] compared the antimicrobials prescribed between 1980 and 1992 in the United States and found an increasing trend in prescribing AX and cephalosporins and a decrease in penicillin prescription. Baggs et al. [41] also found the same trend between 2006 and 2012 among hospitalized patients. This change also has been observed in European populations [4,9], although with significant differences among different regions and countries. The latest publication by Bruyndonckx et al. [4] in 2021 lists the frequency of BLs prescribed in European centers sorting by country from 1997 to 2017, updating the previous ESAC (European Surveillance of Antimicrobial Consumption) studies. It highlights the wide use of AX and AX-CLV, representing 34.8% and 45.9% of patients who received penicillin treatment, which was higher than before. AX-CLV represented more than 50% of total penicillin prescriptions in 11 of 30 countries analyzed. They also described a decrease in BP and other narrow-spectrum penicillin from 15.1% to 10.1% in 2017.

The same situation can be observed with cephalosporins, whose variety and use have increased over time and, therefore, have raised their involvement in hypersensitivity reactions. In our study, cephalosporins were the elicited drugs in 7.7% of cases in Q1, but they became 23.9% in Q4; these were mainly second-generation cephalosporins (11%). Versporten et al. [9] described a similar scenario concerning the use of cephalosporins in a European population, updating also the ESAC studies and listing from 1997 to 2017. Cephalosporin prescriptions represented 11.6% of antibiotic use—similar to previous years—but there was low use of first- and fourth-generation cephalosporins, while maintaining second-generation cephalosporins as the most frequently prescribed (73% of total cephalosporins), especially cefuroxime.

One of the purposes of this study was to analyze whether trend changes in prescribed BLs have influenced the diagnosis of immediate hypersensitivity reactions. In this sense, we have observed that changes in the drugs involved in immediate reactions to BLs have influenced the sensitivity of skin tests. The role in diagnosis played by major and minor determinants of BP has decreased, while that of amoxicillin and, more recently, clavulanic acid and cephalosporins has progressively increased.

Concerning STs results, we observed a decrease over the years in the proportion of patients with positive STs with PPL, MDM, and BP. We detected a decrease from 57.6% of positivity in Q1 to just 22.1% in Q4. In agreement, Blanca et al. [36] described a decrease in the diagnostic sensitivity of PPL, MDM, and BP from 42.1% to 22.1%, compensated by an increase in the sensitivity of AX STs from 26% to 47.5%. Fernandez et al. [5] reported a sensitivity of ST with AX up to 45%. In our group, the positivity of AX STs was 52.8% in the Q4 patients. In the case of cephalosporins, we have previously described different sensitization profiles with different patterns of cross-reactivity and ST results [42]. In order to increase the diagnostic sensitivity of STs, cephalosporins were necessary to be tested for diagnosis, with percentages of positive results around 75% in the latest years. Kahn et al. reviewed cephalosporins ST sensitivity from different studies, finding 72–76% [43].

The change in the proportion of positive results confirms that BP determinants are no longer the most valuable reagents in diagnosing BL hypersensitivity, as proposed by other authors [22,36]. We also found increased sensitivity of STS by adding determinants, especially AX, CLV, and the cephalosporins involved in the reaction, as others have described before [29,37,44]. Currently, there is no commercial MDM and AX available for STs in the United States, making it difficult to standardize studies. STs with cephalosporins also have a limitation for parenteral preparations, mainly in the US [18,43].

These differences in ST results are related to the drugs involved, which vary in prescribing frequency among the groups. The changes in skin testing also reflect the increase in patients with selective reactions, with tolerance to other BLs, as described for AX, CLV, or cephalosporins, and contribute to the increased complexity of the study of BL hypersensitivity reactions [12,45,46]. Performing STs with extended batteries of reagents helped us find selective reactions and search for safe alternatives in our patients.

Despite the specificity of STs, DPT is the gold standard for confirming or excluding the diagnosis, including selective reactions and alternative BL for future use [28]. DPTs were still necessary in 9.8% of our case series, increasing their frequency from 7.6% in Q1 to 14.7% in Q4. Although there were no statistically significant differences among quartiles, the fact reinforces the tendency to increase DPTs because of the decrease in ST sensitivity, which may be due to more available BLs and more different sensitization profiles [5]. Christiansen et al. had a reaction rate of 34.8% when challenging patients with suspected cephalosporins when STS were negative [47], and Bousquet et al. described a similar rate of 30.7% of patients with positive DPTs [48]. Messaad et al. [49] had a lower rate, with 8.4% of patients diagnosed by DPTs, which is closer to ours. These broad differences may be due to the number of performed DPTs and heterogeneity in current practice, as postulated by Romano et al. [29,30]. A total of 6.5% of challenge tests developed an anaphylactic reaction in our patients, but all were mild anaphylaxis. Considering the global sample of 619 patients with confirmed BL hypersensitivity, only 0.65% developed an anaphylactic reaction, which seems to be a safe procedure.

Re-evaluation studies were positive in nearly 9% of patients. These results are similar to those of other groups, with figures varying from 2% to 27.9% [29,50,51]. Differences among quartiles in the number of re-evaluation studies needed, especially when comparing Q1 with the rest, can be related to the extended period between the reaction and the diagnostic study. In Q1, the mean was 161.4 months, while in other groups, means were significantly shorter; the lowest value corresponded to Q4 with 5.39 months (Table 1). These results reinforce the recommended guidelines that patients with old reactions and negative results in a first evaluation should be retested a few weeks later [29,36].

The main limitation of our study is the retrospective nature of the analysis, which might have some heterogeneity in some skin test results; although, it also reflects the constant evolution of allergy tests in beta-lactam hypersensitivity.

## 5. Conclusions

We present a large case series of patients with confirmed immediate reactions to BLs. We can conclude that changes in the pattern of use of some specific BLs in the frequency of prescribing antibiotics might be associated with changes in the sensitization patterns affecting the usefulness and sensitivity of diagnostic procedures.

The evaluation of the diagnostic studies of BL hypersensitivity is a field in constant evolution, which should be continuously monitored and updated to provide the most accurate diagnoses for patients. All countries should implement this monitoring due to the differences in drug use among regions. That also will contribute to classifying the patients better and using more accurate panels of STs, which also help harmonize study protocols. Better use of antibiotics ultimately results in a direct benefit to patients, thus contributing to improved use of health systems.

## Figures and Tables

**Figure 1 biomedicines-10-01535-f001:**
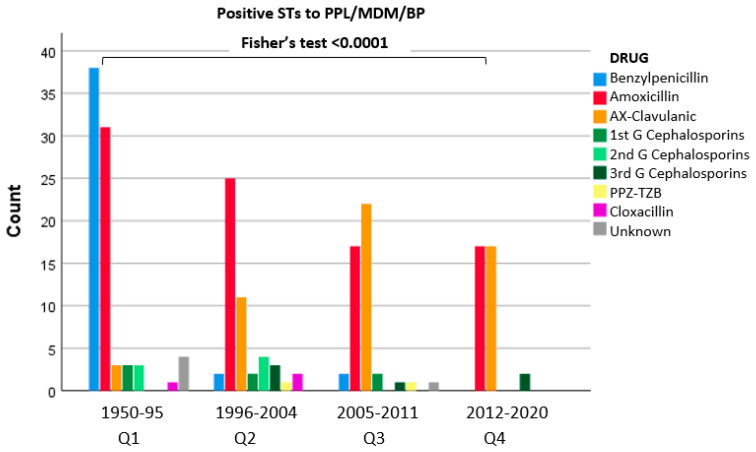
Positive STs with BP reagents related to the eliciting BL.

**Figure 2 biomedicines-10-01535-f002:**
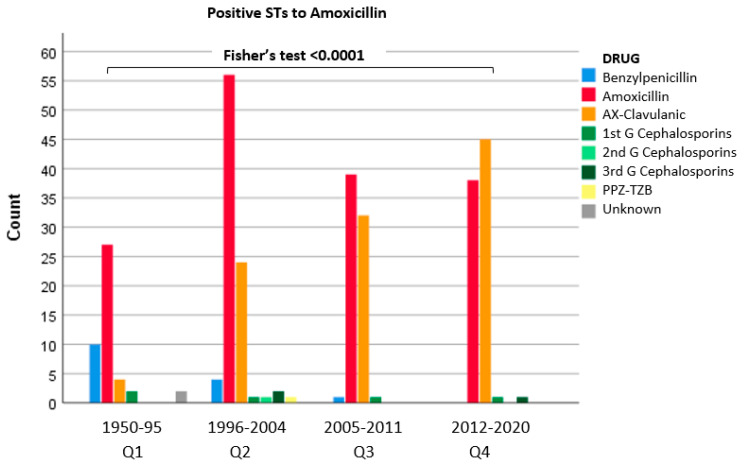
Positive STs with amoxicillin related to the eliciting BL.

**Figure 3 biomedicines-10-01535-f003:**
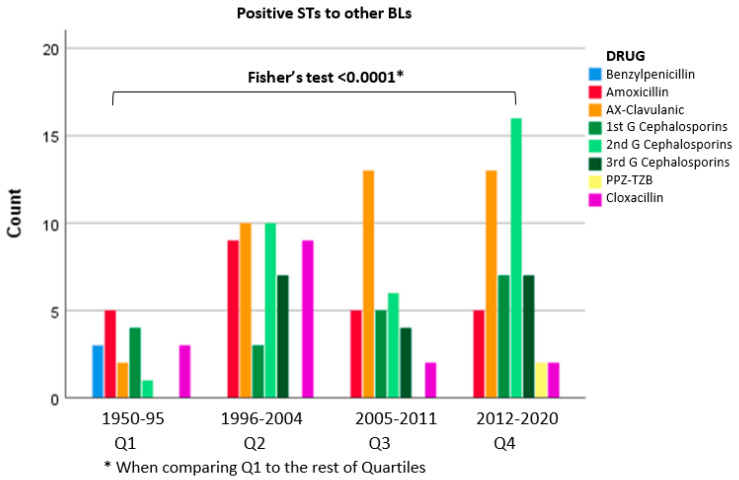
Positive STs to other BLs related to the eliciting BL.

**Table 1 biomedicines-10-01535-t001:** Characteristics of the patients, drug involved, and results of diagnostic procedures.

	Period	Q1 1950–1995	Q2 1996–2004	Q3 2005–2011	Q4 2012–2020	Total	Statistical Significance Difference between Groups
Characteristics							
Patients, n		144	165	144	163	**619**	NA
SEX, n (%)							
Men		64 (44.4)	74 (44.8)	62 (43.1)	74 (45.4)	**274 (44.3)**	n.s
Women		80 (55.6)	91 (55.2)	82 (56.9)	89 (54.6)	**345 (55.7)**	n.s
AGE, n						** n = 526 **	
median ± SD		43.13 ± 16.99	44.12 ± 18.05	46.32 ± 18.34	50.36 ± 17.09	**46.57**	<0.0001
TIME UNTIL STUDY (months)							
mean ± SD		161.43 ± 148.89	23.82 ± 46.17	7.28 ± 12.57	5.39 ± 7.70	**35.5 ± 82.8**	<0.0001 ^1^
REACTION, n (%)						**n = 619**	
Urticaria, angioedema or both		99 (68.8)	91 (55.2)	97 (67.4)	79 (48.5)	**368 (59.5)**	n.s
Anaphylaxis		45 (31.3)	74 (44.8)	47 (32.6)	84 (51.5)	**251 (40.5)**	n.s
IMPLIED DRUG, n (%)						**n = 619**	
BP		48 (33.3)	5 (3)	2 (1.4)	0	**55 (8.9)**	<0.0001
AX		66 (45.8)	75 (45.5)	54 (37.5)	48 (29.4)	**243 (39.4)**	0.007
AX-CLV		7 (4.9)	46 (27.9)	65 (45.1)	70 (42.9)	**188 (30.5)**	<0.0001
First-G. cephalosporins		6 (4.2)	7 (4.2)	6 (4.2)	9 (5.5)	**28 (4.5)**	n.s
Second-G. cephalosporins		5 (3.5)	10 (6.1)	7 (4.9)	18 (11)	**40 (6.5)**	0.038
Third-G. cephalosporins		0	10 (6.1)	6 (4.2)	12 (7.4)	**28 (4.5)**	0.002
PPZ-TZB		0	2 (1.2)	1 (0.7)	4 (1.8)	**7 (1.1)**	n.s
Cloxacillin		3 (2.1)	10 (6.1)	2 (1.4)	2 (1.2)	**17 (2.8)**	0.047
Unknown		9 (6.3)	0	1 (0.7)	0	**10 (1.6)**	<0.0001
DIAGNOSIS, n (%)						**n = 619**	
sIgE		1 (0.6)	1 (0.6)	7 (4.9)	6 (3.8)	**15 (2.4)**	0.032
SPT		28 (17.9)	37 (22.2)	17 (11.2)	23 (14.5)	**105 (17)**	0.051
IDT		84 (53.8)	104 (62.3)	97 (67.4)	101 (63.5)	**385 (62.2)**	n.s
DPT		11 (7.6)	11 (6.7)	13 (9.0)	24 (14.7)	**59 (9.6)**	0.064
Re-SPT		3 (1.9)	6 (3.6)	0	0	**9 (1.5)**	0.016
Re-IDT		26 (16.8)	5 (3.0)	11 (7.6)	6 (3.8)	**43 (6.9)**	<0.0001
Re-DPT		0	2 (1.2)	0	0	**2 (0.3)**	n.s
SKIN TEST RESULTS, n positive (% from the number of patients in the quartile)							
PPL/MDM/BP		83 (57.6)	50 (30.3)	46 (32.8)	36 (22.1)	**215 (34.7)**	<0.0001
AX		45 (31.3)	89 (53.9)	73 (52.5)	85/161 (52.8)	**292 (47.2)**	<0.0001
Others		18 (12.5)	48 (29.1)	35 (24.3)	52 (31.9)	**153 (24.7)**	n.s

BP: benzylpenicillin; AX: amoxicillin; CLV: clavulanic acid; G: generation; PPZ-TZB: piperacillin-tazobactam; SPT: skin prick tests; IDT: intradermal test; DPT: drug provocation test; sIgE: positive specific IgE ImmunoCAP; Re-SPT: re-evaluation skin prick tests; Re-IDT: re-evaluation intradermal tests; Re-DPT: re-evaluation drug provocation tests; NA: not applicable; n.s.: not significant. ^1^: except from Q3 to Q4.

**Table 2 biomedicines-10-01535-t002:** Positive skin test results with the different reagents.

ST Results: n Positive/Performed (%)						
Period Reagent/Drug	Q1 1950–1995	Q2 1996–2004	Q3 2005–2011	Q4 2012–2020	Total	Statistical Significance Difference between Groups
n Patients	144	165	144	163		
**PPL/MDM/BP,** n positive/tested (%)						
BP	38/48 (79.1)	2/5 (40)	2/2 (100)	0/0	42/55 (76.36)	<0.0001
AX	31/66 (47)	25/75 (33.3)	17/52 (32.1)	17/48 (35.4)	90/241 (37.34)	n.s
AX-CLV	3/7 (42.9)	11/46 (23.9)	22/63 (34.9)	17/70 (24.3)	53/186 (28.49)	<0.0001
1st G Cephalosp.	3/6 (50)	2/7 (28.6)	2/6 (33.3)	0/9	7/28 (25)	n.s
2nd G Cephalosp.	3/5 (60)	4/10 (40)	0/7	0/18	7/40 (17.5)	n.s
3rd G Cephalosp.	0/0	3/10 (30)	1/6 (16.6)	2/12 (16.7)	6/28 (21.43)	n.s
PPZ-TZB	0/0	1/2 (50)	1/1 (100)	0/4	2/7 (28.57)	n.s
CLOX	1/3 (33.3)	2/10 (20)	0/2	0/2	3/17 (17.65)	n.s
Unknown	4/9 (44.4)	0/0	1/1 (100)	0/0	5/10 (50)	n.s
**Total**	**83/144 (57.6)**	**50/165 (30.3)**	**46/140 (32.9)**	**36/163 (22.1)**	**215/612 (35.13)**	
**AMOXICILLIN,** n positive/tested (%)						
BP	10/48 (20.8)	4/5 (80)	1/2 (50)	0/0	15/55 (27.3)	<0.0001
AX	27/66 (40.9)	56/75 (74.7)	39/52 (75)	38/47 (80.9)	160/240 (66.7)	n.s
AX-CLV	4/7 (57.1)	24/46 (52.2)	32/61 (52.5)	45/69 (65.2)	105/183 (57.4)	<0.0001
1st G Cephalosp.	2/6 (33.3)	1/7 (14.3)	1/6 (16.7)	1/9 (11.1)	5/28 (17.9)	n.s
2nd G Cephalosp.	0/5	1/10 (10)	0/7	0/18	1/40 (2.5)	n.s
3rd G Cephalosp.	0/0	2/10 (20)	0/6	1/12 (8.3)	3/28 (10.7)	n.s
PPZ-TZB	0/0	1/2 (50)	0/1	0/4	1/7 (14.3)	n.s
CLOX	0/3	0/10	0/2	0/2	0/17	n.s
Unknown	2/9 (22.2)	0/0	0/2	0/0	2/17 (11.8)	0.023
**Total**	**45/144 (31.3)**	**89/165 (53.94)**	**73/139 (52.52)**	**85/161 (52.79)**	**292/608 (48.03)**	
**OTHER DETERMINANTS,** n positive/tested (%)						
BP	3/3 (100)	0/0	0/0	0/0	3/3 (100)	0.001
AX	5/6 (83.3)	9/9 (100)	5/5 (100)	5/5 (100)	24/25 (96)	n.s
AX-CLV	2/2 (100)	10/14 (71.4)	13/14 (92.9)	13/15 (86.7)	38/45 (84.4)	n.s
1st G Cephalosp.	4/4 (100)	3/6 (50)	5/6 (83.3)	7/9 (77.8)	19/25 (76)	n.s
2nd G Cephalosp.	1/4 (25)	10/10 (100)	6/7 (85.7)	16/18 (88.9)	33/39 (84.6)	n.s
3rd G Cephalosp.	0/0	7/9 (77.7)	4/4 (100)	7/11 (63.6)	18/24 (75)	n.s
PPZ-TZB	0/0	0/0	0/0	2/4 (50)	2/4 (50)	n.s
CLOX	3/3 (100)	9/10 (90)	2/2 (100)	2/2 (100)	16/17 (94.1)	0.064
Unknown	NP	NP	NP	NP	NP	
**Total**	**18/22 (81.8)**	**48/58 (82.8)**	**35/38 (92.1)**	**52/64 (81.3)**	**153/182 (84.1)**	

BP: benzylpenicillin; AX: amoxicillin; CLV: clavulanic acid; G: generation; PPZ-TZB: piperacillin-tazobactam; CLOX: cloxacillin; NP: not performed; n.s.: not significant.

**Table 3 biomedicines-10-01535-t003:** Patients with positive STs to implied drug.

Suspected Drug	Reagent Tested	TOTAL Patients n (%)
Benzylpenicillin	PPL/MDM/BP	42/55 (76.36)
Amoxicillin	Amoxicillin	160/240 (66.7)
AX-CLV	Amoxicillin	105/183 (57.4)
	CLV	38/45 (84.4)
1st G Cephalosporins	1st G Cephalosp.	19/25 (76)
2nd G Cephalosporins	2nd G Cephalosp.	33/39 (84.6)
3rd G Cephalosporins	3rd G Cephalosp.	18/24 (75)
PPZ-TZB	PPZ-TZB	2/4 (50)
Cloxacillin	CLOX	16/17 (94.1)

BP: benzylpenicillin; AX: amoxicillin; CLV: clavulanic acid; G: generation; PPZ-TZB: piperacillin-tazobactam; CLOX: cloxacillin.

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
