# Peer review of "Changes in Sensitization Patterns in the Last 25 Years in 619 Patients with Confirmed Diagnoses of Immediate Hypersensitivity Reactions to Beta-Lactams"

_biomedicines, 2022, doi:10.3390/biomedicines10071535_

Round 1

Reviewer 1 Report

Dear Authors,

Congratulation for the manuscript! It is well written!

I have to make only 3 observation:

-Line 95 and line 103: I did not understand the ”allergist's criterion". What did you mean?

-What is 526 in the table? (line 147 table 1).

- I want you to more precisely state that the principal confounder in your article is the change in the pattern of use of some specific beta-lactams, more than the change in the hypersensitivity of a given beta-lactam (if you agree with my sentence!). Please adapt the abstract, discussion and conclusions accordingly! 

Author Response

Dear reviewer: thank you for your review and your comment

Thank you very much for the positive assessment. We are made some amendment over manuscript, trying to answer to all your questions and suggestions.

-Line 95 and line 103: I did not understand the ”allergist's criterion". What did you mean?

We appreciate the question. We meant following the decision of the Allergy specialist in charge of the patient. The manuscript has already been modified including this information.

-What is 526 in the table? (line 147 table 1).

We appreciate the observation. This figure corresponds to the number of patients from which this item was available. And also, with the items “reaction” and “diagnosis.”

- I want you to more precisely state that the principal confounder in your article is the change in the pattern of use of some specific beta-lactams, more than the change in the hypersensitivity of a given beta-lactam (if you agree with my sentence!). Please adapt the abstract, discussion and conclusions accordingly!

We appreciate the comment. Indeed, we agree with the reviewer that there has been a change in prescription patterns of BL antibiotics as reflected in the introduction. Notwithstanding, we believe that the change in the pattern of use of specific beta-lactams might be reflected in changes in sensitization patterns, so we have specifically added to the text that the changes in prescription patterns MIGHT reflect changes in sensitization patterns.

Reviewer 2 Report

Dear Authors,

I read the article with big interest. In particular, I greatly appreciated the attention paid to the trends in prescription influenced sensitization patterns. Before the publication, I would like to ask you to consider following comments:

Material and Methods:
- Please provide details about approval from the Internal Review Board or local Ethical Committee.
- Please confirm that enrolled patients also included minors (from 14 years onwards), and, if so, specify that written informed consent was obtained from parents.
- Drug Provocation Tests (DPTs): please specify whether the protocol adopted (increasing doses starting from 1/8) is local or already present in the literature.
- Statistical analyses: please specify how the distribution of data was evaluated and what was the level of significance chosen.

Results:
- Table 1 (2nd part on page 6 of 13): “2nd G Cephalosp.” is reported two times and “1st G Cephalosp.” is reported three times. Please verify.

References:
Please remove Ref. n° 7. It is a doctoral thesis.

Kind regards

Author Response

We appreciate the reviewer comments and suggestions. Thank you very much for the positive assessment. We are made some amendment over manuscript, trying to answer to all your questions and suggestions.

Below, you can find some clarification, explained after your question.

Material and Methods:
- Please provide details about approval from the Internal Review Board or local Ethical Committee.

We appreciate the comment. As it was a retrospective, real-live study, all informed written consents were approved by the Hospital’s Internal Review Board (line 91).

- Please confirm that enrolled patients also included minors (from 14 years onwards), and, if so, specify that written informed consent was obtained from parents.

Thank you for the observation. Yes, minors were included in the study, and in every case, the parents signed informed consent (lines 106-107).

 - Drug Provocation Tests (DPTs): please specify whether the protocol adopted (increasing doses starting from 1/8) is local or already present in the literature.

We appreciate the comment. Indeed, the dosage of DPTs is a local protocol, as there are not DPT dosage consensus protocols.

- Statistical analyses: please specify how the distribution of data was evaluated and what was the level of significance chosen.

Thank you for the observation. The distribution of data was evaluated with Kolmogorov Smirnov test. Statistical significance was defined as P<.05.

Results:
- Table 1 (2nd part on page 6 of 13): “2nd G Cephalosp.” is reported two times and “1st G Cephalosp.” is reported three times. Please verify.

This was a typo; it is already fixed. Apologize for the mistake.

References:
Please remove Ref. n° 7. It is a doctoral thesis.
We appreciate the observation. Reference number 7 has been removed and changed by another reference.

Round 2

Reviewer 2 Report

Dear Authors,
I have read the revised version of the manuscript based on the Reviewers' comments.
Before the publication, I would like to ask you to consider following comments:

Material and Methods:
- Please provide details about approval from the Internal Review Board. In particular, the Authors should specify: 1) that the study was conducted in accordance with the Declaration of Helsinki, 2) NAME OF INSTITUTE where it was approved, 3) protocol code and date of approval.

Discussion:
- The Authors adopted a local protocol for DPT. Please extend paragraph on DPT by mentioning doi: 10.5414/ALX02104E.

Kind regards

Author Response

Material and Methods:
- Please provide details about approval from the Internal Review Board. In particular, the Authors should specify: 1) that the study was conducted in accordance with the Declaration of Helsinki, 2) NAME OF INSTITUTE where it was approved, 3) protocol code and date of approval.

Thank you for your comment; For retrospective studies, approval code and date are not mandatory. We cannot give information about study protocol; this is a retrospective and observational study, based on our usual clinical practice, so there is no additional intervention on the routine study that patients undergo to make their diagnosis.

The study was carried out following the requirements expressed in the Declaration of Helsinki and the current Spanish legislation on the conduct of observational studies (Royal Decree 957/2020, November 3) and the Law 14/2007 on Biomedical Research. This paragraph has been added to text.

Discussion:
- The Authors adopted a local protocol for DPT. Please extend paragraph on DPT by mentioning doi: 10.5414/ALX02104E.

Thank you for your opinion. We have already done in manuscript.

We hope you find these adequate.

Kind regards.

Round 3

Reviewer 2 Report

The authors have carefully attended to my previous comments.

Author Response

We appreciate the editor comments and suggestions.  We are made some amendment over manuscript, trying to answer to all your questions and suggestions.
Below, you can find some clarification, explained after your question.
1. Line 85: Please revise that the data were collected between 1995 and “2021”.

Thank you for the observation. This was a typo. The data were collected between 1995 and 2020. Apologize for the mistake

2. Table 1:
(1). Please edit Table 1 to make it clear.
Please add “,” before n(%).
Please move “Time until study (months)” to the left side.
Please move “Skin test results, number of positive cases” to the left.
Please add the data of reaction (e.g., angioedema) to Table 1.
Please add the term “Clinical implied” or “Suspected” before DRUG

Thank you for your suggestions. Table 1 has been modified according to your suggestions

3. Please provide an analysis of the relationship between the “clinical implied drug” and the drug with positive skin test in the tested cases.
What is the correlation between the suspected drug and skin test?
Please add a paragraph and a Figure or Table to the manuscript to describe the relationship..

We appreciate the comment. We found a statistically significant correlation between the suspected drug and the drug with positive skin test in the tested cases.

The following paragraph has been added. “When we correlated positive skin test results to different reagents and antibiotics implied in the reactions, we found a statistically significant correlation between positive skin tests with PPL/MDM/BP or AX and reactions related to the administration of BP and amoxicillin, respectively (P<0.0001). Patients presented positive STs to penicillin reagents (PPL/MDM/BP) in 76.3% of patients who suffered reactions with BP, and positive STs to AX in 62.6% of patients in which AX or AX/CLV was the implied drug.  Similarly, a relationship was also found between the cephalosporin involved and the one with positive skin tests”.
We have also included a Table
